# Auditory Brainstem Responses (ABR) of Rats during Experimentally Induced Tinnitus: Literature Review

**DOI:** 10.3390/brainsci10120901

**Published:** 2020-11-24

**Authors:** Ewa Domarecka, Heidi Olze, Agnieszka J. Szczepek

**Affiliations:** 1Department of Otorhinolaryngology, Head and Neck Surgery, Charité-Universitätsmedizin Berlin, Corporate Member of Freie Universität Berlin, Humboldt-Universität zu Berlin, and Berlin Institute of Health, 10117 Berlin, Germany; ewa.domarecka@charite.de (E.D.); Heidi.olze@charite.de (H.O.); 2Faculty of Medicine and Health Sciences, University of Zielona Gora, 65-046 Zielona Gora, Poland

**Keywords:** auditory brainstem response (ABR), tinnitus, animal model, rat

## Abstract

Tinnitus is a subjective phantom sound perceived only by the affected person and a symptom of various auditory and non-auditory conditions. The majority of methods used in clinical and basic research for tinnitus diagnosis are subjective. To better understand tinnitus-associated changes in the auditory system, an objective technique measuring auditory sensitivity—the auditory brainstem responses (ABR)—has been suggested. Therefore, the present review aimed to summarize ABR’s features in a rat model during experimentally induced tinnitus. PubMed, Web of Science, Science Direct, and Scopus databanks were searched using Medical Subject Heading (MeSH) terms: auditory brainstem response, tinnitus, rat. The search identified 344 articles, and 36 of them were selected for the full-text analyses. The experimental protocols and results were evaluated, and the gained knowledge was synthesized. A high level of heterogeneity between the studies was found regarding all assessed areas. The most consistent finding of all studies was a reduction in the ABR wave I amplitude following exposure to noise and salicylate. Simultaneously, animals with salicylate-induced but not noise-induced tinnitus had an increased amplitude of wave IV. Furthermore, the present study identified a need to develop a consensus experimental ABR protocol applied in future tinnitus studies using the rat model.

## 1. Introduction

Tinnitus is an auditory phantom perception despite the absence of external sound. Subjective, chronic tinnitus is diagnosed in humans of all ages, affecting about 16% of the adult population [1,2] and significantly decreasing the life quality of 1–3% of subjects with severe form [3,4,5]. Tinnitus may be a symptom of numerous conditions such as Meniere’s disease, diabetes, arterial hypertension, intracranial hypertension, or hearing loss. The latter, causative reasons, include exposure to noise or ototoxic drugs (i.e., salicylate, cisplatin, quinine) [6,7,8] and have been adapted to induce experimental tinnitus in animals.

A significant challenge during the tinnitus assessment in patients and animal models is the lack of objective diagnostic methods. To date, the clinical evaluation of tinnitus-induced distress is based on the psychometric scores or visual analog scales (VAS) dedicated to measuring various domains. The audiometric assessment of tinnitus (loudness, pitch, residual inhibition) uses the patient’s subjective statement. Similarly, audiometric properties of tinnitus in animals are tested using behavioral paradigms.

There are two established tinnitus induction methods used in animal models: overdose of salicylate and noise exposure [9,10]. Salicylate is known to cause reversible hearing loss and tinnitus in humans [11]. The pioneering work of Jastreboff et al., demonstrating that high doses of salicylate induce tinnitus in animals [10,12], introduced the salicylate-induced tinnitus animal model to basic research. All animals exposed to salicylate develop reversible and dose-dependent pantonal hearing loss co-occurring with tinnitus measured at roughly 16 kHz [9,13]. In contrast to salicylate, noise exposure does not always induce tinnitus. The induction rate varies from 30 to 80%, and the tinnitus frequency depends on the injured part of the auditory periphery [14]. Despite many differences between the salicylate- and noise-induced tinnitus, it is supposed that they share a common mechanistic pathway, converging at the inner hair cell-spiral ganglion neuron synapse [15].

The past decades brought progress in defining the objective, neural correlates of tinnitus. Functional magnetic resonance imaging (fMRI) and electroencephalography (EEG) implicated the association of tinnitus with increased neural synchrony, reorganization of tonotopic maps, and increased spontaneous firing rates (SFR) [9,16]. Clinical studies also demonstrated changes in the auditory brainstem responses (ABR), which are possibly associated with tinnitus [17]. ABR is an early-response auditory evoked potential (AEP). During ABR, the electrical potentials consisting of five waves are isolated from the brainstem’s entire activity response to a calibrated sound. As a result of its objective nature, ABR is used clinically to estimate hearing thresholds of infants, young children, or adult patients who cannot undergo behavioral testing [18,19] and is an essential clinical tool for identifying retrocochlear lesions and vestibular schwannomas [20].

Moreover, ABR is applied in basic animal research to study age-related auditory changes, investigate the effect of therapeutic drugs on auditory potentials, and determine the mechanisms of diseases affecting the auditory system [21,22]. In rats, wave I of ABR associates with the activity of the peripheral auditory nerve. In contrast, waves II–V are thought to originate from the ventral cochlear nucleus, superior olivary complex, lateral lemniscus, inferior colliculus, and medial geniculate body [23]. The first ABR response was demonstrated in two-weeks old rats, whereas the mature ABR response consisting of five waves was observed in five-week-old rats [24].

The ABR amplitudes (II, III, and V wave) provide information about the ABR signal’s natural generators or modulators [25], whereas ABR latency (waves I–V) offers evidence about brainstem function [26]. It is supposed that the reduced amplitude of wave I and an increased hearing threshold might reflect the reduced sensory input found in tinnitus. In agreement with that, salicylate administration and acoustic trauma were found to reduce the ABR amplitude in experimental rats [27]. Nevertheless, the reduced amplitude could also reflect both signal reduction and desynchronization [28]. Therefore, the remaining necessary effort is to determine the ABR features that would indicate the presence of tinnitus independent of hearing loss. In agreement with that, ABR is frequently used in animal studies of tinnitus [29].

The present literature review attempted to extract and synthesize the knowledge about ABR assessment in rats with experimentally induced tinnitus (salicylate administration, noise, and blast exposures). The main goal was to determine whether there is a consistent impact of salicylate, noise, and blast-induced tinnitus on ABR features such as thresholds, latencies, and amplitudes. In addition, changes in the ABR profile of animals with tinnitus were scrutinized in the context of tinnitus-induction methods.

Our work focused on a rat model, and there were several reasons for this decision. The cochlear physiology and anatomy of humans and rats share similarities (e.g., two and a half cochlear turns) compared to guinea pigs (three and a half turns) [30]. Although the highest audible frequencies at 60 dB SPL of humans are 17.6 kHz and of rats 58 to 70  kHz, the lowest audible frequencies are similar (0.03 kHz for humans and 0.52 kHz for rats) [31]. In contrast to mice, early onset of age-related hearing loss (ARHL) has not been reported for rats. Rats can better tolerate noise exposure and rarely exhibit non-inner ear-related symptoms [32,33]. Auditory afferent periphery, believed to be involved during the process of tinnitus generation, has been much studied in rats more extensively than in any other animal model [34]. Importantly, unlike mice, rats do not develop aggressive behavior following noise exposure or salicylate administration [35]; however, they provide an acknowledged model for noise-induced hearing loss [36] and cochlear synaptopathy, which is consistent with hidden hearing loss [37]. Since 1988, rat remains the most prominent species used in tinnitus studies at the behavioral level [38,39]. Moreover, in the pharmacological studies dedicated to developing new tinnitus treatments, rats are a standard animal model [40]. Finally, our laboratory has a long-standing research interest in the effect of stress on the auditory system of rats [25,41,42].

## 2. Materials and Methods

For this study, a literature review was performed in August 2020. A search of the following databanks identified the articles:Science DirectThe Search Engine Tool for Scientific (Scopus)US National Library of Medicine National Institutes of Health (PubMed)Web of Science

The keywords included the following combination of mesh terms:“auditory evoked potential” AND “tinnitus” AND “rat”“auditory brainstem response” AND “tinnitus” AND “rat”“ototoxicity” AND “tinnitus” AND “rat”

Full-text articles were downloaded when the title, abstract, or keywords suggested that the study is eligible for this review. The cardinal eligibility criterion was the publications in the English language. The further procedure of article selection followed the inclusion and exclusion criteria provided below:

Inclusion criteria:Articles published between January 2000 and August 2020Research dedicated to an animal model of tinnitus induced by salicylate administration (When in addition to salicylate, other drugs were applied, only the data related to salicylate were acquired), blast or noise exposure, when the authors used the ABR to measure auditory abilities of animalsUsing ratsOriginal research

Exclusion criteria
Literature review, editorialsFull text not availableArticles not published in English.

The search identified 344 studies. After applying the selection criteria, 36 publications were included in the analysis (Figure 1).

Detailed data extracted from the selected publications are summarized in the Appendix A. The following information was collected:Aim of the articleAge, sex, and strain of ratsSample sizeMethods used to induce tinnitus (salicylate, noise, blast)Methods used to determine the presence of tinnitusStimulus and acquisition characteristics of ABRThe system used to measure ABRSignal intensityRate of signalPolarity of signalThe placement of the electrodesFiltersABR protocolsABR outcome.

The extracted information was analyzed, and the knowledge was synthesized and presented in the Results section.

## 3. Results

### 3.1. Study Selection

Thirty-six studies published between January 2000 and August 2020 met the inclusion criteria. The publication date indicated that during the first decade of the 21st century, the research on that topic was published only sporadically (Figure 2), but the number of publications increased in the second decade.

### 3.2. Strain, Gender, and Age

In the articles selected for this review, Sprague–Dawley rats (both genders) were used in seventeen studies [13,28,44,45,46,47,48,49,50,51,52,53,54,55,56,57]. Wistar rats were used in thirteen [15,27,29,58,59,60,61,62,63,64,65], Long–Evans rats were used in four [66,67,68,69], and Fischer rats (FBN) were used in three studies [49,70,71]. One research group used Fischer 344 and Sprague–Dawley rats [49], two studies used both female and male Wistar rats [29,60], and one article did not provide information about the age of rats [45]. All rats were tested during the first few months of their life. No strain-related differences in the hearing thresholds were found [72,73]. However, compared to Sprague–Dawley rats, Wistar rats develop more aggressive behavior after noise exposure and salicylate administration [74]. Summarized data are presented in Table 1.

In the studies using substance-induced tinnitus, the sample size varied from 4 to 69 animals. Remarkably, only seven of fourteen studies used a control group of animals [27,50,51,58,59,60,66]. In studies using blast- and noise-induced tinnitus, the number of analyzed animals varied from three to 137. Fourteen of the twenty-two studies had a control group [45,48,54,61,62,63,64,65,67,68,69,70,71]. Two articles did not provide information about the experimental and control group [15,45]. In three publications, there was missing data [45,62,63].

### 3.3. Methods Used for Tinnitus Induction

For the induction of tinnitus, three methods were used (Figure 3). In 14 articles, tinnitus was induced with salicylate [13,27,28,29,49,50,51,52,53,58,59,60,66]. Nineteen reports analyzed tinnitus after noise exposure [15,44,45,54,55,56,57,61,62,63,64,65,67,68,69,70,71] and three analyzed tinnitus after blast injury [46,47,48]. These publications were analyzed in one group because of the similarity in the blast- and noise-induced tinnitus mechanisms. They will be referred to as “noise-induced” throughout the rest of this review [75].

### 3.4. Methods Used to Determine the Presence of Tinnitus

The tinnitus-induction rate in rats exposed to salicylate was 100%, whereas the noise exposure resulted in a tinnitus-induction rate that varied from 30 to 80% [14].

Different methods were used to determine if animals have tinnitus. In studies with substance-induced tinnitus, the primary tinnitus-detection method was the gap-prepulse inhibition of the acoustic startle (GPIAS) [13,27,51,52,58,59,60]. GPIAS is based on a hypothesis that assumes that the animals experiencing tinnitus have a fundamental deficit in hearing the silence. Therefore, unlike tinnitus-free animals, animals with tinnitus will not get startled when a sound is played after a short period of silence [44]. In comparison to an operant conditioned procedure, GPIAS does not require training [70]. The operant conditioned procedures require training, e.g., pressing a lever. That type of operant was successfully applied by one study to detect tinnitus [28]. However, six other studies could not determine tinnitus in rats using that method [29,49,50,51,53,66]. Another operant conditioned procedure was “conditioned lick suppression”. During the conditioned leak suppression, the animals choose between the drinking water source: one is a standard bottle, and the other is a spout. Rats are trained to drink from a spout during silence and suppress drinking from a spout during the sound presentation. A light electric shock is used to train the suppressive behavior of rats. If the rats develop tinnitus, they show suppressive behavior and do not use a spout for drinking. This method was used in six studies about noise-induced tinnitus [61,62,63,64,67,68]. In ten further studies, GPIAS was used [44,45,46,47,48,54,55,56,57,69,70]. The last operant-conditioned procedure used to determine tinnitus was a motor task (foraging behavior for sugar water) that was used in three studies [15,65], where animals with tinnitus actively execute the motor task even in the absence of external sound. Rats were trained 3–4 months before the noise exposure [65]. The ratio of activity during an external sound and during periods of silence was used to quantify the motor task [65].

### 3.5. Salicylate-Induced Tinnitus

Salicylate is commonly used to induce tinnitus in rats [38]. In addition to salicylate, quinine or cisplatin were used to study tinnitus in animals [76]. Despite the similarity in inducing tinnitus-like behavior in rats, the mechanisms behind salicylate- and quinine-induced salicylate are different [13]. Salicylate acts by changing the membrane potential and membrane properties [77]. It is a competitive antagonist for the chloride anion-binding site of prestin [78]—a motor protein of outer hair cells [79]. Additionally, salicylate may impact cochlear fast synaptic transmission via the activation of N-methyl-D-aspartate (NMDA) glutamate receptors, accounting for the occurrence of tinnitus [80] and, when administered systemically, it can directly affect the auditory cortex by reversibly depressing the inhibitory, γ-aminobutyric acid (GABA)-ergic neurons [81]. In contrast, quinine acts by altering the cochlear blood flow and the interaction with calcium channels and calcium-dependent potassium channels [77].

Nevertheless, the most commonly used substance in animal studies is a salicylate [38]. A high dose of sodium salicylate induces short-term, reversible tinnitus in rats, as demonstrated by Jastreboff [8,10]. The minimum dose needed to cause tinnitus in rats is 150 mg/kg [12]. Salicylate evokes changes in the peripheral and central auditory system [13,49,52,58]. In the auditory periphery, high doses of salicylate suppress the electromotility of the auditory outher hair cells (OHC) [78]. However, salicylate’s long-term administration increases the prestin expression in the OHCs [49].

In addition, salicylate reduces the amplitude of compound action potentials (CAP) and impacts ABR modulators [82]. The reduced CAP amplitudes in the rats following chronic salicylate treatment indicate long-term functional or structural damage to spiral ganglion neurons (SGN) [49]. The changes are also reflected by the ABR response (reduced amplitude). Nevertheless, the full mechanism of salicylate-induced tinnitus is still unclear.

In the selected studies, both acute and chronic treatments were used (Table 2). Of 14 selected studies, four used ABRs to determine the presence of tinnitus [28,29,50,60]. Two of them focused on developing a diagnostic method by joining ABR with a forward masker of tinnitus. Forward masking results from one stimulus producing temporal inhibition, thus suppressing the subsequent stimulus [28]. Forward masking can create an unmasking effect that results in a recovery of the suppressed ABR and could potentially be used as an objective indicator of tinnitus [29,60]. The method’s usefulness was demonstrated after a single injection of salicylate (300 mg/kg). In four other studies, a correlation between acoustic startle response (ASR) and ABR was examined in rats injected with a single dose of salicylate, 150 mg/kg [50]. In the last study, electrophysiological changes in auditory evoked potentials were analyzed after a 3-day treatment with 350 mg/kg/day [28].

The mechanism of salicylate-induced tinnitus was studied in rats exposed to 200 mg of salicylate/kg/day for three weeks (5 days a week) [49]. The effect of salicylate on ribbon synapses was assessed in rats treated with 200 mg of salicylate/kg/day for ten consecutive days [27]. Changes in the II wave of ABR (representing ventral cochlear nucleus VCN) were examined after four and eight days of salicylate injection at 300 mg/kg/day [58].

### 3.6. Noise-Induced Tinnitus

The degree of noise-induced hearing loss depends on the noise intensity, exposure duration, distance from the noise source, spectrum, and interval length [83]. Only about half of noise-exposed animals developed tinnitus in the research selected for this review [54,64,65]. Despite that, only five of the 22 publications divided the rats into groups with and without tinnitus [15,48,65,67,69]. The effect of noise can be mechanical or metabolic [32]. The mechanical effects of noise include loss of either tip links between hair cells or disruption of actin organization in stereocilia. The metabolic impact involves the generation of oxidative stress that may interfere with hair cell function and neurotransmitter release [32].

#### The Noise Characteristics

The parameters of noise in the animal model are variable. Typically, rats are subjected to high-level noise for 1 to 2 h, binaural or unilateral (when plugged contralateral ear). Mainly, noise characteristics involved 16 kHz, 110–120 dB SPL [45,54,55,56,57,61,62,63,64,67,68]. In three studies, narrow-band noise consisted of 10 kHz and sound level (80–120 dB SPL), and a duration between 1 and 2 h [15,65]. Two researchers’ groups subjected rats for one hour to 116–120 dB SPL and 12 or 17 kHz [44,71].

Binaural exposure to a 10 kHz for 1–2 h led to tinnitus when the sound level was 120 dB SPL but not 80–110 dB SPL [65]. The characteristics of noise used to induce tinnitus are summarized in Table 3.

The frequency used for acoustic exposure impacts the distribution of noise-induced effects along the cochlea and within the brain [84]. The most significant influence of narrow-band noise was often observed above the noise used [32]. The hearing loss accompanying tinnitus was located in the high-frequency range. The hearing loss, hair cell loss, and changes in the central auditory system are less predictable when using broad-band noise than narrow-band noise [38]. Occurred changes also depend on the sound pressure level (SPL) and the experiment duration [85]. It was demonstrated that only rats subjected to 122 dB SPL for two hours demonstrated elevated hearing thresholds (tested: 0.5 to 2 h, 110, 116, and 122 dB SPL, 16 kHz octave band noise (OBN)) [70].

Unlike the others, one research group used more than one unilateral acoustic trauma, the first being 10 kHz, 118–120 dB peak SPL for two hours, and the second (5 weeks later) of identical characteristics, but applied for three hours [69]. In the research regarding blast-induced tinnitus, a single or three consecutive blasts exposures (194 dB), unilateral and bilateral, were used [46,47,48]

ABR in the selected studies was used to assess hearing loss after noise trauma, determine short- and long-term changes in ABR after the noise, and study the efficacy of drugs for hearing loss.

During most studies using noise and blast exposure, rats were anesthetized. However, there were two exceptions [45,69]. In the first work, rats were held in a slowly rotating hardware cloth cage [45,69]. In the second study, rats were subjected to two noise exposures (51). To avoid the protective effect of isoflurane against hearing loss [86], rats were awake during the second exposure (51). Two articles did not provide information about anesthesia during noise exposure [55,68].

### 3.7. Methods of ABR Measurement

#### 3.7.1. ABR Recording Systems

Two commercially available systems—Tucker Davis Technologies (TDT) and Intelligent Hearing Systems (IHS)—were used to record ABR. The majority of the groups used the TDT system [13,27,29,44,45,46,47,48,49,50,51,52,53,54,55,58,59,60,62,66,69,70]. Three research groups used the system [28,51,89], whereas two other groups used both systems [67,68]. Ten articles provided no information about the system used [15,56,57,61,62,63,64,65].

#### 3.7.2. The Stimulus Signal

ABR responses were elicited by click [15,46,47,48,54,65,66,68,69,71] and tone burst stimuli [13,15,27,28,29,44,45,46,48,49,50,51,52,53,54,55,57,58,60,61,62,63,64,65,67,68,69,70,71]. The majority of the research teams used a 3–5 ms tone burst [13,15,28,29,44,45,49,51,53,54,55,57,58,59,60,61,62,64,65,66,67,69,70,71]; five groups applied tone bursts of longer duration, 6–10 ms [46,50,54,68,69]. Fives articles did not provide information about signal [27,48,51,53]. Acoustic stimuli were delivered at <4 kHz [47,65]; 1–45/50 kHz [15,65]; 2.5–40 kHz [45]; 6–16 kHz [29,50]; 8–16 kHz [51]; 4–20 kHz [53]; 4–30 kHz [54]; 2–24 kHz [59]; 4–32 kHz [28,49,66]; 6–20 [44]; 6–32 [28,49,52]; 8–20 kHz or 8–32 kHz [51,55,58,62,63,64,67,70,71]; and 10–32 kHz [56,57]. Sound stimuli were presented at a rate of 10–11 [13,55,65,66], 17–19 [50,53], 20–21 [13,28,44,49,51,55,58,59,63], 29–30 [45,70], or 50 [47,51,56,57,61,62,64,69,71] bursts/s. Eight articles did not provide information about repetition rate [15,27,29,46,48,54,60,68]. Sound stimuli were presented directly to the ear canal [28,44,45,46,48,49,50,51,56,57,58,59,60,61,62,67,69,70] or in free field [13,29,52,53,65,68]. The distance between the speaker and ear were 1 to 10 cm [13,29,52,53,65,68]. Eleven articles provided no information about signal delivering [15,27,47,50,54,55,62,63,64,66,71].

#### 3.7.3. Signal to Noise Ratio and Thresholds

Original data were averaged to achieve an increase of the signal to noise ratio. To estimate thresholds, the number of averageness was 512 [51,58,66] or 1000–1024 [13,28,52]. When the amplitude and latency were analyzed, it was 64–256 [15,65], 200 [60], 300–400 [45,46,47,48,54,69], 512 [53,55,68,70], 600 [49], or 1000–1024 [28,56,57,71]. The sound intensity levels gradually decreased from 110 dB SPL in 5–10 decrements. Five reports did not provide this information to estimate hearing thresholds [44,61,62,63,64]. Click (100 μs) was also used to estimate recovery after noise exposure to calculate the correlation factor (corF) [15,65]. The corF reflects changes in waveform and amplitude before and after noise exposure. High values (around 1) of corF reflect the similarity in a waveform, whereas low values (around 0) indicate loss of both waveform similarity and amplitude [15,65].

#### 3.7.4. Electrode Placement

In the majority of the studies, three stainless-steel recording electrodes were inserted subcutaneously: one on the mastoid of the tested ear (reference electrode), one on the vertex (active), and one on the contralateral mastoid (ground) [13,27,46,47,48,51,52,59,65,69]. Some research groups placed the ground electrode on the back of animals [15,50,62,65,71], on the occiput [61,62,63,64], on the leg [45,67,68,70], or on the nose [29]. One research group placed electrodes on the vertex and ipsi- and contralateral mastoids [54]. In two studies, four electrodes were located at the vertex (active), mastoids (references), and the ground electrode was located on the nose tip [60] or on the back [28]. In four papers, electrodes were placed atypically [53,56,57,58]. Two studies have not provided information regarding electrode placement [44,51].

#### 3.7.5. Filters and Polarity

Only a few publications provided information about filters, polarity, and electrode impedance. Averaged ABR waveforms were bandpass-filtered between 30 Hz and 3 kHz [28]; 100 Hz and 1.5 kHz [51]; 100–3 kHz [13,44,51,52,53,68,70,71]; 10–3 kHz [49]; 300–3 kHz [29,45,46,47,48,54,56,57,60,69,71]; 200 Hz and 5 kHz [65]; or 300–10 kHz [66]. The notch filter was 50 Hz or 60 Hz [29,46,47,48,49,53,60]. The polarity was alternating [49,53,65,66].

### 3.8. Protocols Used for ABR Recordings

#### 3.8.1. Anesthesia

Before the ABR measurement, the rats were anesthetized. The majority of the groups used a mixture of ketamine–HCl and xylazine (ratio: 5:1; 9:1; 1:2:21:1; 6:1; 16:1) [15,27,28,45,48,50,56,57,58,65,66,69,70,71]. In eight publications, isoflurane (4%, 1.5%) was used [13,44,46,48,49,54,67,68]. In addition, chloral hydrate was applied in three cases (0.6 mL/100 g or 400 mg/kg) [29,59,60]. In five studies, instead of xylazine, medetomidine HCl was injected [61,62,63,64]. In one study, a mixture of Zoletil 50 and Rompun 2% or Zoletil (40 mg/kg) + xylazine (10 mg/kg) were used [51,55]. One research group used only ketamine [52], whereas two groups provided no information about the anesthesia [51,68]. In one study, instead of anesthesia, restraint was used [53].

#### 3.8.2. Additional Information

During ABR measurement, animals’ body temperature (37.5 ± 1 °C) was maintained by either a non-heating pad or a warm blanket [28,45,46,47,48,49]. In one study, the body temperature was controlled by maintaining the temperature of 25 °C in a sound-proof room [58]. Even though changes in body temperature modulate brainstem functions [90], most articles have not provided information about maintaining body temperature during measurement [13,15,27,29,44,50,51,52,53,54,55,56,57,59,60,61,62,63,64,65,66,67,68,69,70,71]. Electrode impedance ranged between 1–3 kΩ [29,49,51,53,60].

Detailed information extracted from each publication is presented in Appendix A in the Appendix A.

### 3.9. ABR Evaluation after Salicylate Treatment

The ABR was used to study whether salicylate induces long-term changes in the auditory system [13,28,50,51,52,59,66], to assess the brainstem function following salicylate administration [28,29,50], and to analyze the effect on ABR modulators [27,28,29,49,53,58,60].

#### 3.9.1. Hearing Thresholds after Salicylate Treatment

Exposure to salicylate at 400 mg/kg for seven consecutive days increased hearing thresholds at two tested frequencies (8 and 16 kHz) (on the seventh and eighth day of the experiment). Ten day-long exposure of rats to salicylate at 200 mg/kg/day has not altered the thresholds measured between 2 and 24 kHz compared to the control group [51]. In addition, no changes in hearing thresholds between 6 and 32 kHz were recorded two weeks after the rats’ exposure to salicylate (300 mg/kg) for four consecutive days [52]. The thresholds between 8 and 32 kHz were not affected by a single injection of 350 mg/kg, two hours after drug administration [51]. Studies comparing hearing thresholds during (1–3 days) and after exposure (4–6 days) demonstrated restoring hearing abilities after salicylate cessation. During and two days after the exposure, hearing thresholds were elevated at 4–32 kHz. On the 6th day of the experiment, the threshold shift recovered (elevation at 16 kHz) [28].

Four weeks of exposure to 8 mg of salicylate/mL in water influenced the hearing abilities. A higher mean threshold (at 4–32 dB) was detected in the treated animals compared to the control group, 38 vs. 35 dB, after salicylate cessation [66].

Duron et al. examined changes occurring in auditory function 30–90 min after a single injection of 150 mg/mL in Sprague–Dawley rats (43). Already 1–1.5 h after injection, thresholds were elevated (6–16 kHz). The shift was more remarkable at higher frequencies, increasing by about +8.5 dB at 6 kHz to +17.5 dB at 16 kHz [50].

#### 3.9.2. Effects of Salicylate on ABR Amplitudes and Latencies

The administration of salicylate (300 mg/kg) for eight consecutive days has reduced the ABR amplitudes in all tested frequencies (8–32 kHz, 70 dB) when measured eight days after salicylate cessation. One week later, the reduction of amplitudes lessened [58]. A lower concentration of salicylate (200 mg/kg) applied for ten days has not changed the thresholds, but the reduced amplitude wave I at 2–24 kHz was observed [27]. With a higher concentration of salicylate (350 mg/kg) applied over three days, the amplitude of wave I was reduced, whereas the changes in amplitudes of waves II and IV were seen only on the first day. Wave V was not affected by salicylate [28].

In contrast, wave V’s amplitude was reduced at 12 kHz after a single salicylate injection (250 mg/kg) for at least three days after injection. The amplitude of wave II was also reduced in a broader range (4, 16, and 20 kHz) [53]. Waves IV and I were reduced at 16 kHz 90 min after a single dose (150 mg/kg) [50]. The reduced amplitude of wave III after prolonged exposure to salicylate (3 weeks, five days a week) persisted from 3 days to 4 weeks after cessation [49].

In conclusion, wave I’s amplitude decreased, whereas wave IV’s amplitude increased after salicylate exposure [28,49,50,53,58]. The amplitude of wave III was not affected by a short treatment (1–3 days), but prolonged exposure to salicylate caused the amplitude reduction [28,49,50,58]. The decrease of ABR amplitude was seen even though the threshold was not affected [27]. There was a correlation between the presence of tinnitus and decreased amplitude observed in male Wistar rats. In agreement with that, the return of amplitude to the baseline level correlated with subsidence of tinnitus (based on GPIAS) [58].

A single dose of salicylate (150 mg/kg) affected the latency of wave I and IV. Ninety minutes after salicylate administration, wave I was prolonged at 6, 10, and 12 kHz, whereas the latency of wave IV was reduced (only if measured at 10 kHz) [50]. Intervals III–IV decreased at 6 kHz 60 min after salicylate injection. The changes in latency of wave I were not observed after ten days of exposure to salicylate at the concentration of (200 mg/kg) [27]. The reduced latencies of waves I–IV at 6, 16, and 28 kHz (tested between 4 and 32 kHz) were observed during treatment and one day after salicylate cessation (350 mg/kg/day for three days). On days 2–4th after cessation, only latency II–IV at 32 kHz was prolonged [28]. Changes in latencies were dose-dependent [27,28,50]. Assessment of individual ABR waves revealed alternations dependent on the stimulus frequencies (Table 4).

### 3.10. ABR Evaluation after Noise Exposure

ABR was used to assess the capacity of noise injury to induce the auditory threshold shift [15,44,54,55,56,57,61,62,63,64,65,68,69,70,89]. To estimate the recovery after noise trauma, ABR Waveform Correlation was used [15,65]. Following blast exposure, hearing thresholds and wave I amplitude were analyzed [46,47,48].

#### 3.10.1. Hearing Thresholds after Noise Exposure

Rats subjected to acoustic trauma had elevated hearing thresholds, and the unilateral noise exposure injured only one side. Observed elevated thresholds depended on the noise parameters. Immediately after unilateral noise exposure (16 kHz, 115 dB SPL for 1.5 h), the hearing thresholds in exposed ears (threshold: 70–90 dB peSPL) were higher than in control. Six weeks later, the thresholds were still elevated (70–90 dB peSPL) [56,57]. Four to six weeks after noise exposure, all rats (*n* = 11) presented behavior consistent with tinnitus perception (measured by GPIAS) [57]. Unilateral exposure to noise for one hour (16 kHz, 110 dB) evoked a significant increase in the hearing thresholds at 8–20 kHz in the affected ears (immediately after noise) [61,62]. Simultaneously, five of eight animals tested positive for tinnitus (based on lick suppression task) [62]. Immediately after noise exposure (16 kHz, 116 dB, 1 h), rats had elevated mean threshold (+30–50 dB at 8–32 kHz) [68].

Following noise exposure (8–16 kHz, 115 dB for two hours), the thresholds remained higher for one month in the stimulated ears (threshold: 50–60 dB peSPL), which was not observed in the contralateral ears (thresholds: 20–40 dB peSPL) [54]. After acoustic trauma, four of six rats developed tinnitus (as per GAP detection) [54]. Shorter acoustic exposure (16 kHz, 115 dB, 1 h) induced tinnitus in about 50% of rats (14 animals of 30) tested with a conditioned lick suppression task [64]. The thresholds shift recovered six months after the exposure to noise [63,64].

Long-term hearing loss was observed after a single, unilateral exposure to 120 dB, 16 kHz for one hour. The elevated threshold persisted for 14 months [67]. Although the hearing loss was demonstrated in all rats, not all of them developed tinnitus. Nine months after noise exposure, no differences in thresholds were observed between the noise-exposed rats without tinnitus and the control animals [67]. Restoration to baseline levels after single, unilateral noise (17 kHz, 116 dB for one hour) was observed 16 weeks after noise exposure [71]. At the same time, 10 of 14 noise-exposed rats developed tinnitus at the frequencies between 24 and 32 kHz, as assessed by GAP detection [91].

Rats exposed to 10 kHz, 120 dB for two hours had elevated hearing thresholds at frequencies higher than 8 kHz in exposed ears for at least 15 days following noise exposure (frequencies tested: 1–50 kHz) [15]. While there were no differences in hearing thresholds between tinnitus-positive and -negative animals, rats with lesser tinnitus had a slighter reduction in the number of synapses on inner hair cells [15]. The difference in hearing thresholds in rats with and without tinnitus was described by Rüttiger et al. [65]. Rats with noise-induced tinnitus (binaural exposure for one hour, or one and a half hours, 10 kHz, 120 dB SPL) had a significantly larger hearing loss than rats without tinnitus. Six days after one-hour noise exposure, the hearing loss in frequencies above 11.3 kHz was considerably higher than the tinnitus-free group. Thirty days after 1.5 h noise exposure, an elevated hearing threshold in animals with tinnitus was also significantly greater at low frequencies (tested: 1–45 kHz). Tinnitus was determined in five of 15 and five of 17 rats, respectively (the motor task) [65]. The lower intensity of the above noise (<120 dB, 10 kHz, 1–2 h) did not evoke elevation in hearing thresholds in Wistar rats [65].

The selected studies detected different patterns of audiometric changes following noise exposure. Two hours after single unilateral noise exposure (12 kHz,120 dB, one-hour exposure), ABR thresholds increased by about 30–45 dB at 12 kHz (tested range: 6–20 kHz). While some rats developed permanent hearing loss (>2 weeks), other animals’ hearing abilities recovered within three consecutive days. The changes were absent in contralateral ears [44]. The observed differences might be related to the used outbred rat strain (Sprague–Dawley). A different pattern was observed in other studies when acoustic exposure of 16 kHz, 116 dB for two hours, was used [45]. After one week, an increase in hearing thresholds between 2.5 and 40 kHz was observed between SPL 10 and 40 dB [45].

Turner et al. demonstrated that noise with an intensity below 122 dB SPL (16 kHz, 0.5–2 h) did not evoke changes in the hearing thresholds two hours after exposure (8–32 kHz) [70]. That observation was confirmed by the study of Kim et al. Despite extended noise exposure time (four hours), the next day, no elevated hearing thresholds (8–32 kHz) were determined in the experimental animals [55].

The effect of two exposures on behavior consistent with tinnitus perception was demonstrated in Long–Evans rats. Following 1 to 8 weeks after two noise exposures (10 kHz, 118–120 dB, 2 and 3 h), rats with tinnitus had broader hearing loss than rats without tinnitus (8–28 kHz vs. 12–28 kHz, respectively). Tinnitus was demonstrated in 12 of 18 rats (GPIAS) [69].

Bilateral blast (194 dB for 10 ms) produced an immediate elevation in hearing thresholds from 39 to about 62 dB peSPL at frequencies less than 4 kHz [47]. On days 14, 28, and 90, after the blast, thresholds recovered to 37.86 dB SPL, 30.71 dB SPL, and 27.86 dB SPL, respectively [47].

In another study, 14 days after a single unilateral blast, the hearing level of animals returned to baseline [47]. In contrast, Sprague–Dawley rats, following a unilateral blast, had elevated hearing thresholds between eight and 28 kHz in both ears one day after blast exposure. Three to six weeks later, threshold elevation was still observed but only in the frequencies between 16 and 28 kHz and only in the blast-exposed ears [46].

In the selected studies, not all animals developed tinnitus. Unfortunately, only a few studies performed audiometric analyses of the animals with positive behavioral tests indicative of tinnitus [65,67]. These analyses demonstrated differences in the hearing thresholds between the animals with and without tinnitus. After the noise exposure, rats with tinnitus had increased hearing thresholds [65] and a broader hearing loss than the noise-exposed animals without tinnitus [69]. Rats with tinnitus had elevated thresholds at 8–28 kHz, whereas rats without tinnitus had elevated thresholds at 12–28 kHz (following two noise exposures). Nevertheless, there was no correlation between threshold shift and GPIAS results [69]. In disagreement with the above results, Brozoski et al. [67] observed no differences in hearing thresholds between rats with and without tinnitus.

#### 3.10.2. Effects of Noise on ABR Amplitudes and Latencies

The influence of noise exposure on the alternation and recovery of the ABR waveform was analyzed by calculating the click-induced ABR waveform correlation factor (CorF) [15,65]. The CorF reflects changes in waveform and amplitude [65] before and after noise exposure. High values (around 1) of corF are indicative of similarities in waveforms, whereas low values (about 0) indicate loss of both waveform similarity and amplitude [15,65]. Fifteen days after single noise exposure (10 kHz,120 dB, two hours, unilateral), the overall ABR amplitude was reduced. Simultaneously, the ABR threshold was elevated at frequencies higher than 8 kHz (assessment of 1–50 kHz) [15]. Moreover, a negative relationship between tinnitus and a number of synaptic contacts on the inner hair cells was observed (tinnitus test: the motor task), as reflected by the ABR wave I [15].

While the ABR waveform of rats exposed to noise (10 kHz, 110 dB for two hours) was restored within two weeks, rats exposed to higher noise intensity (120 dB) had distorted ABR waveforms and reduced amplitude two weeks after noise trauma [65]. The overall amplitude reduction was observed in rats 6–30 days after bilateral noise exposure (10 kHz, 120 dB, 1 or 1.5 h) [65]. Compared to tinnitus-free rats, rats with tinnitus had a reduced hearing recovery 6 and 30 days after noise trauma [65]

After the unilateral blast, the threshold recovered fully within five weeks (8–28 kHz), whereas the amplitude of wave I at 28 kHz was still reduced [48]. Tinnitus was determined in eight of 13 animals (GPIAS). There were no differences in ABR between tinnitus and tinnitus-free rats, and the ABR amplitudes were reduced in all exposed rats [48]. In summary, noise exposure induces ABR amplitude reduction. Table 5 summarizes the observed changes.

Some common similarities determined in ABR amplitudes after salicylate and noise exposure are presented in Figure 4.

## 4. Discussion

This review aimed to evaluate the auditory brainstem response changes occurring in rats’ auditory systems after experimentally induced-tinnitus. Thirty-six studies published between January 2000 and August 2020 and dedicated to the rat model met the inclusion criteria. Based on the selected studies, we summarize the knowledge of audiometric changes occurring in tinnitus using ABR.

In the selected articles, the principal inducers of tinnitus (noise and salicylate) were used. Rats exposed to salicylate had changes in hearing thresholds and ABR waveforms in specific frequencies [49,50,51,53,59]. Despite considerable differences between the studies, two consistent parallels were identified. The first similar finding was a consistent reduction in wave I and increased wave IV (Figure 4). The reduction of the wave’s I amplitude reflects changes in sensory input. Although there was no significant hair cell loss, a decrease in ribbon synapses was observed [27,59]. In addition, abnormalities in both presynaptic elements and postsynaptic nerve fibers were observed [27]. Reduced sensory input could have lead to the enhanced auditory midbrain responses such as the increased amplitude of wave IV, reflecting changes in the inferior colliculus [65]. In contrast to the salicylate treatment, the noise has induced a reduction in the amplitude of wave IV [15]. The imbalance in excitation and inhibition occurring on the level of inferior colliculus might contribute to tinnitus development [92].

The relation between behavior consistent with tinnitus and the ABR changes was investigated using the female Sprague–Dawley rats exposed to salicylate (350 mg/kg/day for three days, gavage administration) [28]. There was a correlation between tinnitus behavior and changes in ABR amplitude (waves II–V) and latency (waves II and III) [28]. Such association was observed only for specific frequencies (4, 8, 16, and 32 kHz). Nevertheless, upon the return of amplitudes to the baseline level, tinnitus behavior disappeared. There was no association between the amplitude and latency of wave I and hearing thresholds [28]. Fang et al. observed that ABR amplitude was reduced when cochlear sensitivity improved (increased amplitude of the distortion product otoacoustic emissions—DPOAE) [58].

While salicylate evoked tinnitus in all rats, only half of them demonstrated tinnitus after noise exposure [14]. A higher hearing threshold and amplitude reduction were observed in rats subjected to acoustic trauma. After single noise exposure (16 kHz, 120 dB, 1.5 h, bilateral) and two noise exposures (10 kHz, 118–120 dB peSPL, two hours; then five weeks later for three hours, unilateral), rats with tinnitus had higher hearing thresholds than tinnitus-free animals [65,69]. This difference was not observed in rats exposed to 16 kHz, 120 dB for one hour [67]. The variance seems to be an effect of different noise parameters and various times of ABR assessment.

Some differences were also observed regarding the auditory inner hair cells (IHC) synaptic contacts in rats with and without tinnitus [15]. After noise exposure (10 kHz, 120 dB for 1 or 1.5 h), a greater reduction of ribbon synapses in basal and mid-basal turn was observed in rats with tinnitus [65]. No loss of ribbon synapses was seen in the cochlear apical turn after salicylate treatment or noise trauma [60,65]. Cochlear deafferentation depends on the degree of inner hair cell synaptopathy. Two of the studies included in the present review confirmed the notion about the loss of IHCs ribbon synapses (deafferentation), leading to tinnitus when the ABR was reduced. Upon the restoration of ABR, tinnitus was no longer observed [65], indicating that the degree of IHC ribbon loss might be a crucial factor for the recovery of ABR after acoustic trauma and tinnitus generation [65].

In one article, restraint was used during ABR measurement [53]. Restraint applied for four hours was previously shown to impact animals’ hearing abilities by inducing stress and the hypothalamus–pituitary–adrenal (HPA) axis, thus increasing the systemic levels of corticosterone, which protected animals from the acoustic trauma [89]. Rats subjected to restraint demonstrated abnormalities in CAP and DPOAE. Interestingly, another type of stress—social stress—induced an increase in IHC ribbon synapses [65]. Furthermore, repeated injections might also be a source of anxiety for animals [93]. In the process of data evaluation, we compared hearing thresholds at baseline between mock-injected and non-injected animals and found that the injected rats had higher hearing thresholds by about 10–20 dB [61,62,63,64], which has so far not been reported.

It has been suggested that tinnitus can internally mask the ABR [94]. Therefore, ABR with a forward masker is supposed to be an objective indicator of tinnitus and was used after salicylate treatment in female and male Wistar rats. The forward masker and the probe were presented to both ears. After the above stimulation, rats had a reduction in ABR amplitude and prolonged latencies (I–V). This was not observed in rats treated with salicylate [29,60]. Additional experiments with a larger sample size should be performed to address this exciting issue.

The data extracted and analyzed in the present review suggest that a loss of cochlear inner hair cell ribbon synapses could contribute to the development of tinnitus reflected by reducing the amplitude of wave I, which was also observed in human studies [17]. Some studies suggest measuring abnormalities in wave V’s latency to indicate cochlear synaptopathy in humans after noise exposure [95]. The term “cochlear synaptopathy” was proposed to describe damage at the cochlear synapse without any loss of hair cells, resulting in “hidden hearing loss” [96].

The eardrum perforation is one of the most frequent injuries after blast exposure [91]. Despite the middle ear’s known impact on ABR response, none of the articles dedicated to the blast injury provided information about the middle ear status after blast [46,47,48].

The factors that could affect ABR are summarized in Table 6. It is our recommendation to use this summary as a template for the experimental outlines.

Several limitations were identified in the data collected. The first limitation was a lack of precise information about the experimental protocol during ABR measurements such as animals’ body temperature or anesthesia. Both of these factors could influence the ABR recordings. Each decrement at 0.5 °C of the body temperature may significantly alter ABR latencies and amplitudes [105,106]. What is more, rats anaesthetized with isoflurane (1.5–2%; 4%) have higher hearing thresholds in comparison to thresholds after an administration of ketamine + xylazine (50 mg/kg + 9 mg/kg) [101,103]. One study reported a lack of amplitude differences after blast exposure between the rats anesthetized with isoflurane (4%) or ketamine. However, other researchers observed poorer ABR responses after isoflurane (dose not known) [48,70]. What is more, there are no studies on the impact of anesthesia on substance-induced tinnitus.

The second limitation was a variation in the frequencies tested by ABR. While the standard testing range includes <4 to 24/32 kHz, some selected papers provide information only for single frequencies [97].

The third pitfall is various techniques used for ABR recording, such as signal delivery. Acoustic stimuli were delivered directly to the ear or in the free field. These different conditions may also contribute to the differences in latencies and amplitudes reported.

The fourth limitation was a high sample size variation, which might have impacted the results (e.g., reduction, improvement vs. no changes in amplitude of wave II, III, and V) in rats following salicylate [28,49,50,53,58]. Nevertheless, the sample size was not used as an exclusion criterion because of our review’s character.

The fifth limitation is not reporting whether the gender differences influenced rats’ hearing abilities [29,60]. Older male rats (8–24 months old) had higher ABR thresholds and overall smaller amplitude than female rats. Despite the lack of ABR differences between young female and male rats, adult female rats had shorter latencies (I–IV) than the male rats [100,107]. Various studies also demonstrated a link between hearing thresholds and the menstrual cycle [111,112], suggesting that the estrous cycle phase should be considered when using female rats.

The sixth limitation is the variety of methods used to determine the presence of tinnitus in rats.

The studies using ABR for the evaluation of experimentally induced tinnitus in the rat model demonstrated differences in hearing abilities after tinnitus induction. Observed changes depended on the drug dose, noise intensity, time point of audiometric measurement, and frequency used during ABR measurement. Differences in results were also observed even if the experiments were conducted by the same research group [27,59].

All the above limitations indicate a great need to create a universal protocol, at least for each research group, if not for all of them. Nevertheless, the most consistent finding across all studies was a general reduction of ABR amplitudes in the animals experiencing noise-induced tinnitus. In contrast, In rats with salicylate-induced tinnitus, the amplitude of wave I was also reduced, but the amplitude of wave IV increased.

Clinical studies in tinnitus patients with normal hearing (frequency 0.25–8 kHz) demonstrated reduced amplitude of wave I at high intensities (80–90 dB SPL). The amplitude of wave V was not affected compared to the control group [113]. Despite the reduction of wave I, a normal wave V, might be an effect of increased neural responsiveness in the central auditory system to compensate for the reduced activity of the auditory nerve [113]. No differences in the amplitude of wave V in tinnitus patients and an average hearing threshold were also described in the study of Kehrle et al. [114]. In contrast, Gu et al. observed a higher amplitude of wave V in patients with tinnitus [115]. The authors suggested that wave’s V higher amplitude is an artifact induced by a lower frequency filter cutoff [115]. To sum up, ABR amplitude changes were determined in patients with tinnitus and normal hearing thresholds (based on pure tone audiometry). Reduction in the amplitude of wave I likely indicates a cochlear synaptopathy, whereas the unchanged or elevated amplitude of wave V could reflect central regions’ compensated responses [17].

Studies in patients with tinnitus and high-frequency hearing loss demonstrated greater amplitude of wave III than in the control group without tinnitus (threshold-, sex-, and age-matched) [116]. Interestingly, such differences were not observed in a previous study published by the same group. Nevertheless, the mean ABR amplitudes tended to be reduced [117].

In 2017, a review dedicated to studying changes in the ABR of patients with tinnitus was published [17]. The results indicated a high level of heterogeneity between the clinical studies. This heterogeneity was attributed to different etiologies of tinnitus, gender, age, and various protocols used for ABR recording. Similar diverseness was observed in our review. Interestingly, similar to ABR’s study in individuals with tinnitus, animal studies’ most consistent finding was a reduced amplitude of wave I. Despite the similar methods for tinnitus induction in animal studies, there is still a considerable heterogeneity of results suggesting a possible intrinsic heterogeneity of tinnitus and the importance of using a standardized universal protocol to perform the experiments.

## 5. Future Directions

For future studies, we acknowledge using the rat model as offering three significant advantages: (i) the presence of similarities in the phantom character of tinnitus between rats and humans; (ii) the ability to perform the audiometric measurements before and after tinnitus onset; and (iii) the possibility to control and modify experimental factors influencing tinnitus. However, in future investigations, sufficient details regarding the tinnitus induction and ABR protocols (factors listed in Table 6) should ensure the data reproducibility and facilitate future work.

## 6. Conclusions

The publications included in the present review provided audiometric characteristics of hearing loss and contributed to a better understanding of tinnitus in the rat model. In animals with salicylate-induced tinnitus, ABR waves had a shorter latency as in the control animals, reduced amplitude of wave I, and increased amplitude of wave IV. In contrast, in animals with noise-induced tinnitus, all ABR waves had reduced amplitudes, implicating that salicylate and noise induce different changes in the auditory brainstem which still result in the tinnitus percept. Contrasting changes in the amplitude of wave IV could reflect fluctuations in the auditory processing that are not related to tinnitus but might be associated with the cause of tinnitus. The overall evidence collected in the present work urges to establish a universal protocol for electrophysiological recording and animal handling in future animal studies on tinnitus.

## Figures and Tables

**Figure 1 brainsci-10-00901-f001:**
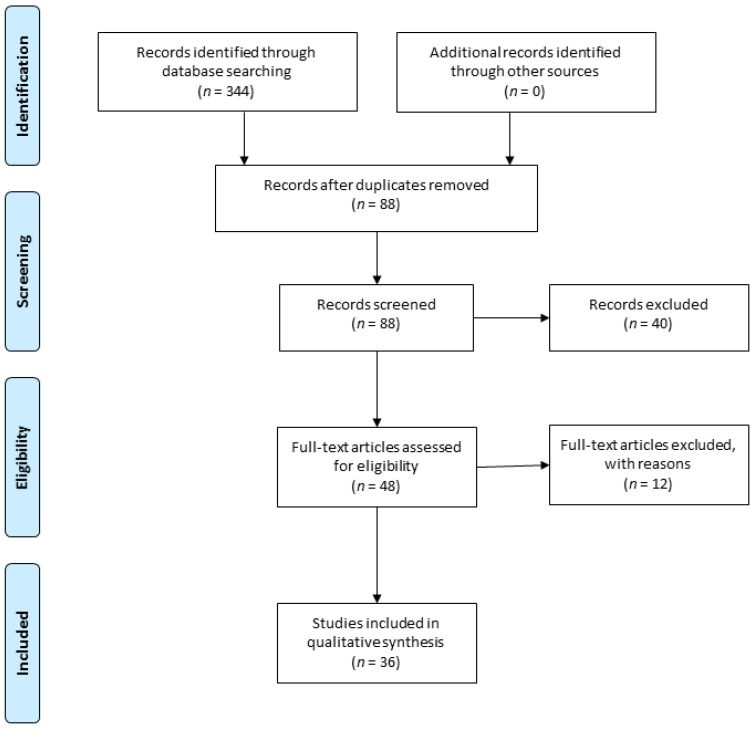
PRISMA diagram of the study selection process [43]. “*n*” signifies the number of publications.

**Figure 2 brainsci-10-00901-f002:**
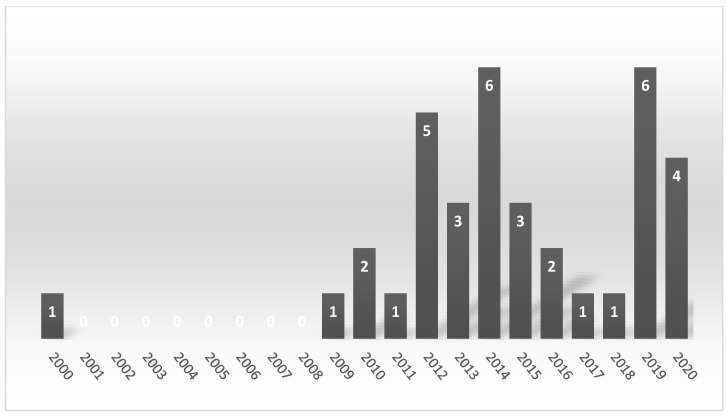
The number of publications regarding auditory brainstem responses (ABR) and rat model of tinnitus per year.

**Figure 3 brainsci-10-00901-f003:**
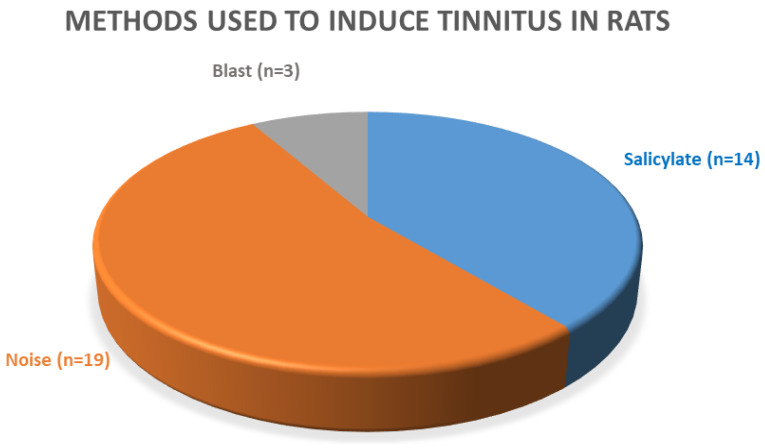
Methods used to induce tinnitus in the selected articles (*n* = 36).

**Figure 4 brainsci-10-00901-f004:**
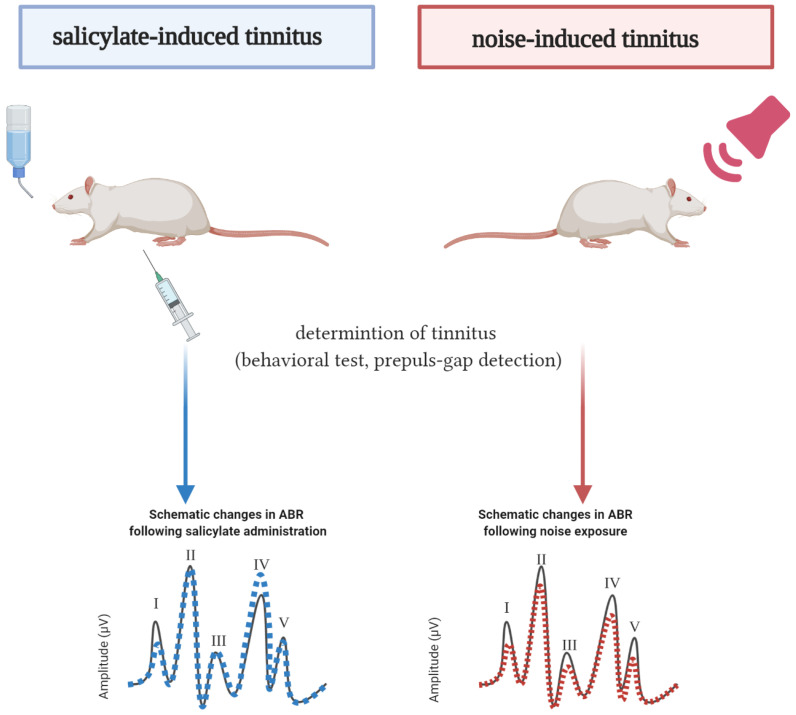
Schematic representation of the ABR profile’s similarities and differences between the control rats and animals with tinnitus (created with BioRender.com). Of 36 publications included in this review, four publications regarding noise-induced tinnitus and six publications regarding salicylate-induced tinnitus qualified for the quantitative ABR analyses. The solid black line represents the ABR of the control animals. The dashed blue line represents the ABR of rats with salicylate-induced tinnitus [27,28,49,53,58,59]. The red dashed line represents the ABR of rats with noise-induced tinnitus [15,65]. Only the changes consistently reported by two or more studies were taken into account. The changes in latencies are not presented due to inconsistencies between the studies.

**Table 1 brainsci-10-00901-t001:** Rat strain, gender, and age.

Strain	Sprague–Dawley (Used in 17 Studies)		Wistar (Used in 13 Studies)			Long-Evans (Used in 4 Studies)		Fischer FBN; 344 (Used in 3 Studies)	
gender	Male (used in 16 studies)	Female (used in 1 study)	Male (used in 8 studies)	Female (used in 3 studies)	Female and male (used in 2 studies)	Male (used in 4 studies)	Female (no publications	Male (used in 2 studies)	Female (no publications)
age (range)	1.5–4 months	1.5–5 months	1.5–5 months	2–5 months		2–3 months	-	3–6 months old	

**Table 2 brainsci-10-00901-t002:** Features of experiments with salicylate-induced tinnitus.

Article	Sodium Salicylate Concentration	Frequency of Salicylate Application	Lenght of Salicylate Exposure	Method of Application	Tinnitus Measurement Method
Bauer et al., 2000 [66]	8 mg/mL	4 weeks	4 weeks	orally (in drinking water)	No assessment
Chen et al., 2010 [49]	200 mg/kg	Once a day	5 days per week for 3 weeks	injection	No assessment
Zhang et al., 2020 [27,59]	200 mg/kg	Once a day	10 consecutive days	intraperitoneal injection	GPIAS
Fang et al., 2016 [58]	300 mg/kg	Once a day	8 consecutive days	intraperitoneal injection	GPIAS
Jang et al., 2019 [51]	400 mg/kg	Once a day	7 consecutive days	intraperitoneal injection	GPIAS
Ralli et al., 2010, 2014 [13,52]	300 mg/kg	Once a day	4 consecutive days	injection	GPIAS
Castañeda et al., 2019 [28]	350 mg/kg	Once a day	3 consecutive days	orally (by gavage)	an operant conditioned-suppression procedure
Lee et al., 2019 [51]	350 mg/kg	single application	single application	intraperitoneal injection	No assessment
Liu and Chen, 2015 [60]	300 mg/kg	single application	single application	intraperitoneal injection	GPIAS
Liu and Chen, 2012 [29]	300 mg/kg	single application	single application	injection	No assessment
Sawka and Wei, 2014 [53]	250 mg/kg	single application	single application	intraperitoneal injection	No assessment
Duron et al., 2020 [50]	150 mg/kg	single application	single application	intraperitoneal injection	No assessment

**Table 3 brainsci-10-00901-t003:** Features of experiments with noise-induced tinnitus.

Article	Laterality of Noise Application	Noise Intensity	Noise Duration	Anesthetic Used during Noise Exposure	Determination of Tinnitus in Rats/Sample Size	Timepoint of Tinnitus Determination
Kim et al., 2020 [55]	Bilateral	16 kHz, 112 dB SPL	4 h	Data not available	Data not available (GPIAS)	One day after the noise and 1 and 10 days after completing DEX administration
Brozoski et al., 2019 [67]	Unilateral (contralateral ear-plugged)	16 kHz, 120 dB SPL	1 h	Isoflurane (1.7%)	Not all (an operant conditioned-suppression procedure)	3 and 9 months after noise
van Zwieten et al., 2019 [57]	Unilateral (contralateral ear-plugged)	16 kHz, 115 dB SPL	1.5 h	Ketamine (90 mg/kg) + Xylazine (10 mg/kg)	11/11 (GPIAS)	4–6 weeks after noise
van Zwieten et al., 2019 [56]	Unilateral (contralateral ear-plugged)	16 kHz, 115 dB SPL	1.5 h	Ketamine (90 mg/kg) + Xylazine (10 mg/kg)	(Data not available GPIAS)	4–6 weeks after noise
Ahsan et al., 2018 [54]	Unilateral (contralateral ear-plugged)	8–16 kHz, 115 dB	2 h	Isoflurane (2–3%)	4/6 (GPIAS)	After noise (no details)
Turner and Larsen, 2016 [70]	Unilateral (contralateral ear-not announced)	16 kHz, 110, 116 or 122 dB SPL; 8 or 32 kHz or BBN, 110 dB SPL	0.5 h, 1 h, or 2 h	Ketamine + Xylazine (doses—not announced)	Data not available (GPIAS)	On Day 1, 3, 7, 14, 21, 28 after noise exposure and monthly thereafter over until 1 year
Bing et al., 2015 [15]	Unilateral (contralateral ear-plugged)	10 kHz, 120 dB	2 h	Medetomidine hydrochloride (0.33 mg/kg)	Data not available (the motor task)	3 and 10 days after noise
Zheng et al., 2015 [64]	Unilateral (contralateral ear-plugged)	16 kHz, 115 dB	1 h	Fentanyl (0.2 mg/kg) + Medetomidine hydrochloride (0.5 mg/kg)	14/30 (a conditioned lick suppression task)	1 month after noise
Zheng, McPherson and Smith, 2014 [63]	Unilateral (contralateral ear-plugged)	16 kHz, 115 dB,	1 h	Medetomidine hydrochloride (0.33 mg/kg)	Data not available (a conditioned lick suppression task)	2 weeks and then at 10 and 17.5 weeks after noise
Laundrie and Sun, 2014 [44]	Unilateral (contralateral ear-plugged)	12 kHz, 120 dB SPL	1 h	Isoflurane (1–2%)	Data not available GPIAS)	4 h after noise
Ropp et al., 2014 [45]	Unilateral (contralateral ear-plugged)	16 kHz, 116 dB SPL	2 h	Unanesthetized (rat was held in a slowly rotating hardware cloth cage)	Data not available (GPIAS)	At various delays after noise (2–3 times a week for 1–4 months)
Rüttiger et al., 2013 [87]	Binaural	10 kHz, 120 dB SPL	1 h or 1.5 h	Ketamine (75 mg/kg) + Xylazine hydrochloride (5 mg/kg)	5/15 and 5/17. (the motor task)	Before and at 6 day (1 h) or 30 days (1.5 h) after noise
Pace and Zhang, 2013 [69]	Unilateral (contralateral ear-plugged)	10 kHz, 118–120 dB peak SPL	Two hours, five weeks later, the 2nd exposure for 3 h	First: Isoflurane (5%); second: while rats awake	12/18 (GPIAS)	One day after the 1st noise and two times a week until six weeks after the 2nd noise
Singer et al., 2013 [65]	Binaural	10 kHz, 80,100, 110, or 120 dB SPL	1–2 h	Ketamine hydrochloride (75 mg/kg) + Xylazine hydrohloride (5 mg/kg)	Only rats subjected to 120 dB demonstrated tinnitus (the motor task)	6–14 days after noise exposure
Brozoski et al., 2012 [68]	Unilateral (contralateral ear-plugged)	16 kHz, 116 dB	1 h	Data not available	Data not available (an operant conditioned-suppression)	Immediately after noise
Zheng et al., 2012 [62]	Unilateral (contralateral ear-plugged)	16 kHz, 110 dB	1 h	Ketamine hydrochloride (75 mg/kg) + Medetomidine hydrochloride (0.3 mg/kg)	5/8 (a conditioned lick suppression task)	After noise (no details)
Zheng et al., 2012 [62]	Unilateral (contralateral ear-plugged)	16 kHz, 110 dB	1 h	Ketamine hydrochloride (75 mg/kg) + Medetomidine hydrochloride (0.3 mg/kg)	Data not available (a conditioned lick suppression task)	Two weeks after noise
Zheng et al., 2011 [61]	Unilateral (contralateral ear-plugged)	16 kHz, 110 dB	1 h	Ketamine hydrochloride (75 mg/kg) + Medetomidine hydrochloride (0.3 mg/kg)	Data not available (a conditioned lick suppression task)	2 weeks and 10 months after noise
Wang et al., 2009 [88]	Unilateral (contralateral ear-plugged)	17 kHz, 116 dB SPL	1 h	Ketamine hydrochloride (50 mg/kg) + Xylazine (9 mg/kg)	10/14 (GPIAS)	20 days after noise every 2 weeks up to 16 weeks
Ouyang 2017 [48]	Unilateral (contralateral ear-plugged)	194 dB SPL	Single blast exposure	Isoflurane (4%) or Ketamine (100 mg/kg) + Xylazine (10 mg/kg)	8/13 (GPIAS)	After blast (2 times per week)
Mahmood et al., 2014 [46]	Unilateral (contralateral ear-plugged)	Data not available	3 consecutive blast exposure	Isoflurane (3%)	Data not available (GPIAS)	1 h after the last blast and for 8 weeks afterward
Mao et al., 2012 [47]	Bilateral	194 dB SPL	Single blast exposure (10 ms)	Ketamine (100 mg/kg) + Xylazine (10 mg/kg)	Data not available (GPIAS)	1, 14, 28, and 90 days after blast

**Table 4 brainsci-10-00901-t004:** Salicylate-induced changes in ABR amplitude and latency.

Article	SAL Dose	Rats	Wave I	Wave II	Wave III	Wave IV	Wave V	Intervals	Threshold
Zhang et al., 2020 [27,59]	200 mg/kg/day per 10 days	Male, Wis	↓ __	Not tested	Not tested	Not tested	Not tested	__	__ (vs. control group)
Fang et al., 2015 [58]	300 mg/kg per 8 days	male, Wis	Not tested	Not tested	↓	Not tested	Not tested	Not tested	Not tested
Duron et al., 2020 [50]	150 mg/kg/day	male, SD	↓ →	__	__	↑ ←	Not tested	III–IV: ←	↑ at 6–16 kHz
Castañeda et al., 2019 [28]	350 mg/kg per 3 days	female, SD	↓	↑	__	↑	__	I–IV: ← II–IV: →	↑ at 4–32 kHz
Sawka and Wei, 2014 [53]	250 mg/kg/day	male, SD	Not tested	↓	Not tested	Not tested	↓	Not tested	Not tested
Chen et al., 2010 [49]	200/mg/kg/day for three weeks (5 days per week)	male, SD	Not tested	Not tested	↓	Not tested	Not tested	Not tested	Not tested

↓ reduced; ↑ increased; __ no changes; → prolonged; ← reduced; SD Sprague–Dawley; Wis Wistar.

**Table 5 brainsci-10-00901-t005:** Changes in amplitude in rats after noise exposure.

Article	Noise Details	Rats	Wave I	Wave II	Wave III	Wave IV	Wave V	Intervals	Threshold
Bing et al., 2015 [15] *	10 kHz, 120 dB, 2 h, unilateral	female, Wis	↓	Not tested	Not tested	↓	Not tested	Not tested	↑ at 8–50 kHz
Singer et al., 2013 [65]	10 kHz, 120 dB SPL for 1–2 h, binaural	female, Wis	↓	↓	↓	↓	↓	Not tested	↑ at 2–4 kHz
Rüttiger et al., 2013 [65]	10 kHz, 120 dB SPL, 1 or 1.5 h, binaural	female, Wis	↓	↓	↓	↓	↓	Not tested	↑ at 1–45 kHz
Ouyang et al., 2017 [48]	blast exposure (194 dB SPL), unilateral	male, SD	↓	Not tested	Not tested	Not tested	Not tested	Not tested	_

↓ reduced; ↑ increased; _ no changes; → prolonged; ←reduced; Wis Wistar rats; SD Sprague–Dawley rats * Despite calculations of corF (which is done with the values of waves I-V), only the amplitudes of wave I and wave IV were provided [15].

**Table 6 brainsci-10-00901-t006:** Factors possibly influencing ABR response.

Factor	Influence on ABR	Suggestions Based on ABR User Guide [97]	Additional Recommendations
Experimental area	Cables, noise generators might generate electrical noise	A sound attenuating chamber with a built-in Faraday cage	Before starting experiments, conduct a saline test (to determine the noise floor)
Speaker placement	Affect the stimulus level	The speaker should be on the same plane as the tested ear and set at an angle from the sides of the enclosure	Place the speaker in a distance of 10 cm away from the animal
Electrode placement	Incorrect recording	Vertex (active); reference (ipsilateral ear); ground (contralateral ear or hind hip or base of tail)	Write the lot number of electrodes in the protocol
Electrode impedance	<4 kΩ, lower artifacts suppression, low quality of ABR recording	1 k–3 kΩ	
Repetition rate	If the repetition rate increases, latency increases, too [88,98] With increased stimulus rates (click, 5/s to 50/s), aged Fischer 344 rats demonstrated an increase in latencies wave IV and V and overall amplitude reduction (I–V) [99] A similar result was observed in young Sprague–Dawley rats in response to click and pure tones [100] Increasing the repetition rate shortens the recovery time, but it also reduces ABR measurement times and shortens the time of anesthesia [87]	21/s	The rate 21/s minimizes the effects of noise from the 50/60 Hz cycle of mains power (95)
Hearing range tested		Standard Testing Range: >4 kHz to 14/32 kHz	
Type of stimulus	Data not available	Click: 100 μs, pure tone: 5 ms (2-1-2)	-
Number of averages	Impact on signal/noise ratio. Rats with hearing loss require more averages than rats with normal hearing	512 averages	512 ensures a balance between the signal quality and minimalization of the time to complete testing
Polarity	Data not available	Alternating	-
Anesthesia	Rats following isoflurane (1.5–2%; 4%) administration have higher hearing thresholds in comparison to ketamine + xylazine (50 mg/kg + 9 mg/kg) [101,102] In addition, isoflurane significantly impairs DPOAEs, whereas a normal dose of ketamine + xylazine (50 mg/kg + 10 mg/kg) in Sprague–Dawley rats does not [102,103]	A mixture of ketamine + xylazine (a weight-dependent mg/kg dosage) Ketamine can be used to keep the subject anesthetized longer than 45 min	Monitor rats under anesthesia
Tympanic membrane evaluation	Properly functioning ossicular chain conduction is a prerequisite for ABR recording	Evaluate tympanic membrane using otoscope before the start of the experiment	See picture in [104]
Body temperature of rats	Temperature decreasing by 0.5 °C or more degrees may significantly alter ABR latencies and amplitudes [105,106]	Heating pads should be used to maintain body temperature (37 °C) or control the temperature in the experimental room (25 °C)	Monitor the body temperature with a rectal probe throughout the recording
Gender	There were no significant gender-dependent differences in amplitudes or latencies between the ages of 14 and 70 days in Sprague–Dawley rats [100] However, adult female rats had shorter latencies (I–IV) than male rats [100,107]	-	In female rats >5 weeks old, the estrous cycle should be controlled [108]
Age	Immature auditory response of rat pups [109] Age-related effects on hearing	-	Write the age and the body weight in the protocol
Strain	The hearing range of laboratory subjects varies across different strains (95)		
Day-night cycle	The sensitivity to noise varies at different daytimes. Two weeks after noise trauma (during the night 9 PM) ABR thresholds were elevated, whereas in mice exposed to noise at 9 AM (6–12 kHz, 100 dB SPL for 1 h) [110]	-	Note the time of experiments
Handling rats	Handling is a well-known source of stress-induced variation in animal studies [98]	-	
Housing rats	Disruption in factors below evokes stress reactions in rats, which mediates hearing abilities in rats. Factors: Maintaining a stable temperature, humidity, and light–dark cycle in the facility, free access to water and chow	-	Maintaining a stable temperature and humidity in the facility; Standard chow and water ad libitum; acclimatization one week before running the experiment; providing an enriched environment

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
