# Peer review of "Auditory Brainstem Responses (ABR) of Rats during Experimentally Induced Tinnitus: Literature Review"

_brainsci, 2020, doi:10.3390/brainsci10120901_

Round 1

Reviewer 1 Report

The paper reviews a very important topic relative to tinnitus, i.e. its correlations with objective tests in model animals.

General remarks:

1-The implications are stated as a bit too strong (i.e. line 20 - The data extracted confirm that in rats, ABR might reflect tinnitus- related modifications). There is so much variability among studies, and with hearing loss also a confounding factor, I would hesitate to attribute ABR changes after treatments to tinnitus presence. Please state review results in a more neutral way.

2-Authors should clearly state why the rat has been chosen for this review above other animal models (e.g. mouse, guinea pig) which have also been used. A little background comparison would help (e.g. species-related synaptopathy differences) in the introduction.

3-Since all drug-related discussion has been centered upon salicylate, the studies mentioning other drugs as tinnitus inducers should be left out of the paper entirely, as they are very few and do not add any insight.

4- Please update and/or integrate clinical studies; the initial description is a bit sloppy/incomplete/outdated.

5- English is sometimes not clear. Please have the paper checked by a language proofreader.

6- Figures are somewhat amateurish.

More in detail, comments follow relative to lines in the paper:

L40: behavioural (in animals) and subjective (in human) are not in contrast, but similar

L46: there is a review on clinical use of ABR in tinnitus (Milloy et al,. 2017; quoted as ref.14). Please refer to those results and to the review’s discussion (which also quote the paper in ref.11 but put it in a larger context: quoting ref.11 only in here would be biased).

L50: paper (ref.12) refers to rats only, so better talk about that rather than rodents in general. Actually, are there differences between Wistar and other rat strains?

L56: development of clinical and research electrophysiology is parallel, so “was adapted to” appears unjustified

L68: this hypothesis is only one of the many. Time course is very different for noise trauma and salicylate- induced tinnitus. Moreover, salicylate effects are found at several levels in the auditory system, and focusing on one only seems unjustified. In rats, salicylate also induces hearing loss: how do you separate the two effects?

L73: reduced amplitude may reflect both signal reduction and desynchronization. Please elaborate.

L180 explain procedure better

L236 what is the bandwidth?

L248 why 16 kHz? Wasn’t it broadband?

L331- comparison between results of different studies is unclear for the whole salicylate section, and also (although less) for noise traumas. Please better separate results from different studies, so that contrasting or concordant results are more evident

L429 outbred

L467 what is the meaning of (tested: ) ?

L505 wave II is not decreased in the figure

L508 reduced OR DESYNCHRONIZED sensory input

L535: unclear. When ABR is back to normal do ribbons reappear?

L544 explain protocol better

L600. Add that in animal studies, despite the similar methods for tinnitus induction, there is still a large heterogeneity. Therefore, either methods are extremely critical or tinnitus is intrinsically heterogeneous. Several studies come from the same research groups. Are results within a group more consistent than overall?

It would be useful to recommend or sketch guidelines for tinnitus induction with noise or drugs in the rat. What needs to be recorded? What needs to be better standardized?

Author Response

The authors thank the Reviewer for the time and thoughts spent on this work. We are grateful for the constructive criticism.

General remarks:

1-The implications are stated as a bit too strong (i.e. line 20 - The data extracted confirm that in rats, ABR might reflect tinnitus- related modifications). There is so much variability among studies, and with hearing loss also a confounding factor, I would hesitate to attribute ABR changes after treatments to tinnitus presence. Please state review results in a more neutral way.

We mildened up our statements and used neutral statements, e.g., lines 655-657: “In contrast, in animals with noise-induced tinnitus, all ABR waves had reduced amplitudes, implicating that salicylate and noise induce different changes in the auditory brainstem which still result in the tinnitus percept.”

2-Authors should clearly state why the rat has been chosen for this review above other animal models (e.g. mouse, guinea pig) which have also been used. A little background comparison would help (e.g. species-related synaptopathy differences) in the introduction.

A new paragraph was inserted in the introduction:

“Our work focused on a rat model, and there were several reasons for this decision. Cochlear physiology and anatomy of humans and rats share similarities (e.g., two and a half cochlear turns) compared to guinea pigs (three and a half turns) [28]. Although the highest audible frequencies at 60 dB SPL of humans are 17.6 kHz and of rats 58 to 70  kHz, the lowest audible frequencies are similar (0.03 kHz for humans and 0.52 kHz for rats) [29]. In contrast to mice, early onset of age-related hearing loss (ARHL) has not been reported for rats. Rats can better tolerate noise exposure and rarely exhibit non-inner ear-related symptoms [30, 31]. Auditory afferent periphery, believed to be involved during the process of tinnitus generation, has been much studied in rats more extensively than in any other animal model [32]. Importantly, unlike mice, rats do not develop aggressive behavior following noise exposure or salicylate administration [33]; however, they provide an acknowledged model for noise-induced hearing loss [34] and cochlear synaptopathy, consistent with hidden hearing loss [35]. Since 1988, rat remains the most prominent species used in tinnitus studies at the behavioral level [36, 37]. Moreover, in the pharmacological studies dedicated to developing new tinnitus treatments, rats are a standard animal model [38]. Finally, our laboratory has a long-standing research interest in the effect of stress on the auditory system of rats [23, 39, 40].”

We also discuss the synaptopathy later in the Discussion:

“Some differences were also observed regarding the IHC synaptic contacts in rats with and without tinnitus [15]. After noise exposure (10 kHz, 120 dB for one or 1.5 hours), a greater reduction of ribbon synapses in basal and mid-basal turn was observed in rats with tinnitus [60]. No loss of ribbon synapses was seen in the cochlear apical turn after salicylate treatment or noise trauma [55, 60]. Cochlear deafferentation depends on the degree of inner hair cell synaptopathy. Two of the studies included in the present review confirmed the notion about the loss of IHCs ribbon synapses (deafferentation), leading to tinnitus when ABR was reduced. Upon restoration of ABR, tinnitus was no longer observed [60], indicating that the degree of IHC ribbon loss might be a crucial factor for the recovery of ABR after acoustic trauma and tinnitus generation [60].”

3-Since all drug-related discussion has been centered upon salicylate, the studies mentioning other drugs as tinnitus inducers should be left out of the paper entirely, as they are very few and do not add any insight.

We followed the Reviewer's suggestion and removed one study not dealing with salicylate but left the second, as it was partially investigating the effects of salicylate.

4- Please update and/or integrate clinical studies; the initial description is a bit sloppy/incomplete/outdated.

We modified the section dealing with clinical studies and it reads now as follows:

“Clinical studies in tinnitus patients with normal hearing (frequency 0.25–8 kHz) demonstrated reduced amplitude of wave I at high intensities (80-90 dB SPL). The amplitude of wave V was not affected compared to the control group [111]. Despite the reduction of wave I, a normal wave V, might be an effect of increased neural responsiveness in the central auditory system to compensate for the reduced activity of the auditory nerve [111]. No differences in amplitude of wave V in tinnitus patients and an average hearing threshold were also described in the study of Kehrle et al. [112]. In contrast, Gu et al. observed a higher amplitude of wave V in patients with tinnitus [113]. The authors suggested that wave V's higher amplitude is an artifact induced by a lower frequency filter cutoff [113]. To sum up, ABR amplitude changes were determined in patients with tinnitus and normal hearing thresholds (based on pure tone audiometry). Reduction in the amplitude of wave I likely indicates a cochlear synaptopathy, whereas the unchanged or elevated amplitude of wave V could reflect central regions' compensated responses [17].

Studies in patients with tinnitus and high-frequency hearing loss demonstrated greater amplitude of wave III than in the control group without tinnitus (threshold-, sex-, and age-matched) [114]. Interestingly, such differences were not observed in a previous study published by the same group. Nevertheless, the mean ABR amplitudes tended to be reduced [115].

In 2017, a review dedicated to studying changes in ABR of patients with tinnitus was published [17]. The results indicated a high level of heterogeneity between the clinical studies. This heterogeneity was attributed to different etiologies of tinnitus, gender, age, and various protocols used for ABR recording. Similar diverseness was observed in our review. Interestingly, similar to ABR's study in individuals with tinnitus, animal studies' most consistent finding was a reduced amplitude of wave I. Despite the similar methods for tinnitus induction in animal studies, there is still a considerable heterogeneity of results suggesting possible intrinsic heterogeneity of tinnitus and the importance of using a standardized universal protocol to perform the experiments.”

5- English is sometimes not clear. Please have the paper checked by a language proofreader.

The manuscript was revised accordingly.

6- Figures are somewhat amateurish.

Figures were done in a professional fashion. We changed Figure 1 and used the official PRISMA template. Figure 4 was also replaced with a hopefully better version.

More in detail, comments follow relative to lines in the paper:

L40: behavioural (in animals) and subjective (in human) are not in contrast, but similar

That mistake was corrected.

L46: there is a review on clinical use of ABR in tinnitus (Milloy et al,. 2017; quoted as ref.14). Please refer to those results and to the review’s discussion (which also quote the paper in ref.11 but put it in a larger context: quoting ref.11 only in here would be biased).

We revised that paragraph, and it reads now as follows:

“The past decades brought progress in defining the objective, neural correlates of tinnitus. Functional magnetic resonance imaging (fMRI) and electroencephalography (EEG) implicated the association of tinnitus with increased neural synchrony, reorganization of tonotopic maps, and increased spontaneous firing rates (SFR) [9, 16]. Clinical studies also demonstrated changes in the auditory brainstem responses (ABR), possibly associated with tinnitus [17]. ABR is an early-response auditory evoked potential (AEP). During ABR, the electrical potentials consisting of five waves are isolated from the brainstem's entire activity response to a calibrated sound. Because of its objective nature, ABR is used clinically to estimate hearing thresholds of infants, young children, or adult patients who cannot undergo behavioral testing [18, 19] and is an essential clinical tool for identifying retrocochlear lesions and vestibular schwannomas [20].”

L50: paper (ref.12) refers to rats only, so better talk about that rather than rodents in general. Actually, are there differences between Wistar and other rat strains?

We revised the sentence and actually the entire paragraph accordingly.

“Moreover, ABR is applied in basic animal research to study age-related auditory changes, investigate the effect of therapeutic drugs on auditory potentials, and determine the mechanisms of diseases affecting the auditory system [21, 22]. In rats, wave I of ABR associates with the activity of the peripheral auditory nerve. In contrast, waves II-V are thought to originate from the ventral cochlear nucleus, superior olivary complex, lateral lemniscus, inferior colliculus, and medial geniculate body [23]. The first ABR response was demonstrated in two-weeks old rats, whereas the mature ABR response consisting of five waves was observed in five-week-old rats [24].”

We appreciate the question - the differences in steady-state ABR and generally in hearing abilities between Wistar and other rat strains are a subject of our intense research, and we hope to publish on this topic soon.

L56: development of clinical and research electrophysiology is parallel, so “was adapted to” appears unjustified

Right. The sentence reads now:

“Moreover, ABR is applied in basic animal research to study age-related auditory changes, investigate the effect of therapeutic drugs on auditory potentials, and determine the mechanisms of diseases affecting the auditory system [21, 22].”

L68: this hypothesis is only one of the many. Time course is very different for noise trauma and salicylate- induced tinnitus. Moreover, salicylate effects are found at several levels in the auditory system, and focusing on one only seems unjustified. In rats, salicylate also induces hearing loss: how do you separate the two effects?

Truth - in that paragraph, the focus is on the introduction of the two systems, and it is not our intention to get into discussion on the origin of tinnitus. The sentence reads now:

“Despite many differences between the salicylate- and noise-induced tinnitus, it is supposed that they share a common mechanistic pathway, converging at the inner hair cell-spiral ganglion neuron synapse [15].”

L73: reduced amplitude may reflect both signal reduction and desynchronization. Please elaborate.

We revised this statement and it reads now:

“It is supposed that the reduced amplitude of wave I and an increased hearing threshold might reflect the reduced sensory input found in tinnitus. In agreement with that, salicylate administration and acoustic trauma were found to reduce the ABR amplitude in experimental rats [27]. Nevertheless, the reduced amplitude could also reflect both signal reduction and desynchronization [28]. Therefore, the remaining necessary effort is to determine the ABR features that would indicate the presence of tinnitus independent of hearing loss. In agreement with that, ABR is frequently used in animal studies of tinnitus [29].”

L180 explain procedure better

We added the following:

“During the conditioned leak suppression, the animals choose between the drinking water source: one is a standard bottle, and the other is a spout. Rats are trained to drink from a spout during silence and suppress drinking from a spout during the sound presentation. A light electric shock is used to train the suppressive behavior of rats. If the rats develop tinnitus, they show suppressive behavior and do not use a spout for drinking.

L236 what is the bandwidth?

Bandwidth is the difference between the upper and lower frequencies. We removed this expression as it is confusing.

L248 why 16 kHz? Wasn’t it broadband?

We apologize but could not find a mention of 16 kHz in line 248 or the neighboring lines.

L331- comparison between results of different studies is unclear for the whole salicylate section, and also (although less) for noise traumas. Please better separate results from different studies, so that contrasting or concordant results are more evident

The entire manuscript was revised and restructured. The essential details are represented in Table 4, 6 and 6. 

L429 outbred

Corrected.

L467 what is the meaning of (tested: ) ?

Corrected. The (tested: ) was removed and the sentence revised. It reads now:

“Some differences were also observed regarding the IHC synaptic contacts in rats with and without tinnitus [15]. After noise exposure (10 kHz, 120 dB for one or 1.5 hours), a greater reduction of ribbon synapses in basal and mid-basal turn was observed in rats with tinnitus [65].”

L505 wave II is not decreased in the figure

We have produced a new Figure 4 and used the “order of appearance“ of the tinnitus induction, meaning salicylate first and noise second. In that new figure, all curves have been double-checked and represent a consensus of at least two or more publications.

L508 reduced OR DESYNCHRONIZED sensory input

We revised that paragraph and it reads now:

“The reduction of the wave's I amplitude reflects changes in sensory input. Although there was no significant hair cell loss, a decrease in ribbon synapses was observed [27, 59]. Also, abnormalities in both presynaptic elements and postsynaptic nerve fibers were observed [27]. Reduced sensory input could have lead to the enhanced auditory midbrain responses such as the increased amplitude of wave IV, reflecting changes in the inferior colliculus [65]. In contrast to the salicylate treatment, the noise has induced a reduction in the amplitude of wave IV [15]. The imbalance in excitation and inhibition occurring on the level of inferior colliculus might contribute to tinnitus development [91].”

L535: unclear. When ABR is back to normal do ribbons reappear?

Not quite – the authors of cited references suggest that other factors (e.g. Arc mobilization ), which were not in the focus of this review, might play a role in that process.

Here a quote from that paper:

“We observed that IHC ribbon loss (deafferentation) leads to tinnitus when ABR functions remain reduced and Arc is not mobilized in the hippocampal CA1 and AC. If, however, ABR waves are functionally restored and Arc is mobilized, tinnitus does not occur. Both central response patterns were found to be independent of a profound threshold loss and could be shifted by the corticosterone level at the time of trauma.”

L544 explain protocol better

All details of the protocols are listed in Supplementary data.

L600. Add that in animal studies, despite the similar methods for tinnitus induction, there is still a large heterogeneity. Therefore, either methods are extremely critical or tinnitus is intrinsically heterogeneous. Several studies come from the same research groups. Are results within a group more consistent than overall?

We added the following sentence at the end of Discussion:

Despite the similar methods for tinnitus induction in animal studies, there is still a considerable heterogeneity of results suggesting possible intrinsic heterogeneity of tinnitus and the importance of using a standardized universal protocol to perform the experiments.

The results from one group are actually not as consistent as it could be expected.

It would be useful to recommend or sketch guidelines for tinnitus induction with noise or drugs in the rat. What needs to be recorded? What needs to be better standardized?

We used the existing Table 6 and expanded the information, and added recommendations.

Reviewer 2 Report

This review focuses on the literature where auditory brain stem responses (ABRs) have been used to assess hearing in animals during experimentally induced tinnitus. The studies using rats as an animal model were included in this review. This work also emphasizes on couple recent papers where ABRs have been suggested as an objective method for tinnitus assessment. Although this review is interesting and cover all targeted literature, the rationale behind it and the focus on one animal is not clear. Based on the abstract, one of the main reasons to pay attention to ABRs in the tinnitus research is that this method can be potentially used for tinnitus assessment. This is a highly misleading statement. There are only a couple of studies which suggest that, but this hypothesis has never been tested in these studies directly. Therefore, it is highly premature to recommend ABRs to be such a technique and to make it as one of the main reasons for this review. The focus on the rat animal model is also surprising and not well justified. Many other tinnitus animal models have been used and therefore it is not clear why we need to pay attention to ABRs in rats but not in other models.  

Author Response

The authors thank the Reviewer for the time and thoughts spent on this work. We are grateful for the constructive criticism.

This review focuses on the literature where auditory brain stem responses (ABRs) have been used to assess hearing in animals during experimentally induced tinnitus. The studies using rats as an animal model were included in this review. This work also emphasizes on couple recent papers where ABRs have been suggested as an objective method for tinnitus assessment. Although this review is interesting and cover all targeted literature, the rationale behind it and the focus on one animal is not clear. Based on the abstract, one of the main reasons to pay attention to ABRs in the tinnitus research is that this method can be potentially used for tinnitus assessment. This is a highly misleading statement. There are only a couple of studies which suggest that, but this hypothesis has never been tested in these studies directly.

Thank you for pointing this out. We have changed the entire abstract. It is written in a more cautios way, suggesting a possible association of certain ABR features with processes influenced by tinnitus.

Abstract: Tinnitus is a subjective phantom sound perceived only by the affected person and a symptom of various auditory and non-auditory conditions. The majority of methods used in clinical and basic research for tinnitus diagnosis are subjective. To better understand tinnitus-associated changes in the auditory system, an objective technique measuring auditory sensitivity – the auditory brainstem responses (ABR) - has been suggested. Therefore, the present review aimed to summarize ABR's features in a rat model during experimentally-induced tinnitus. PubMed, Web of Science, Science Direct, and Scopus databanks were searched using MESH terms: auditory brainstem response, tinnitus, animal model, rat. The search identified 344 articles, and 36 of them were selected for the full-text analyses. The experimental protocols and results were evaluated, and the gained knowledge was synthesized. A high level of heterogeneity between the studies was found regarding all assessed areas. The most consistent finding of all studies was reduced ABR wave's I amplitude following exposure to noise and salicylate. Simultaneously, animals with salicylate- but not noise-induced tinnitus had an increased amplitude of wave IV. Furthermore, the present study identified a need to develop a consensus experimental ABR protocol applied in future tinnitus studies using the rat model.”

Therefore, it is highly premature to recommend ABRs to be such a technique and to make it as one of the main reasons for this review.

The entire manuscript has been revised. Our intention was to recommend caution and controlled use of ABR when performing the animal research.

The focus on the rat animal model is also surprising and not well justified. Many other tinnitus animal models have been used and therefore it is not clear why we need to pay attention to ABRs in rats but not in other models.  

We followed this suggestion and introduced a clear justification for our focus on a rat model:

“Our work focused on a rat model, and there were several reasons for this decision. Cochlear physiology and anatomy of humans and rats share similarities (e.g., two and a half cochlear turns) compared to guinea pigs (three and a half turns) [30]. Although the highest audible frequencies at 60 dB SPL of humans are 17.6 kHz and of rats 58 to 70  kHz, the lowest audible frequencies are similar (0.03 kHz for humans and 0.52 kHz for rats) [31]. In contrast to mice, early onset of age-related hearing loss (ARHL) has not been reported for rats. Rats can better tolerate noise exposure and rarely exhibit non-inner ear-related symptoms [32, 33]. Auditory afferent periphery, believed to be involved during the process of tinnitus generation, has been much studied in rats more extensively than in any other animal model [34]. Importantly, unlike mice, rats do not develop aggressive behavior following noise exposure or salicylate administration [35]; however, they provide an acknowledged model for noise-induced hearing loss [36] and cochlear synaptopathy, consistent with hidden hearing loss [37]. Since 1988, rat remains the most prominent species used in tinnitus studies at the behavioral level [38, 39]. Moreover, in the pharmacological studies dedicated to developing new tinnitus treatments, rats are a standard animal model [40]. Finally, our laboratory has a long-standing research interest in the effect of stress on the auditory system of rats [25, 41, 42].”

Round 2

Reviewer 2 Report

Thank you for addressing all my concerns.